# Effect of Temperature, Syngas Space Velocity and Catalyst Stability of Co-Mn/CNT Bimetallic Catalyst on Fischer Tropsch Synthesis Performance

Omid Akbarzadeh [1],*, Solhe F. Alshahateet [2], Noor Asmawati Mohd Zabidi [3], Seyedehmaryam Moosavi [4], Amir Kordijazi [5], Arman Amani Babadi [1], Nor Aliya Hamizi [1], Yasmin Abdul Wahab [1], Zaira Zaman Chowdhury [1] and Suresh Sagadevan [1],*

1 Nanotechnology & Catalysis Research Centre, University of Malaya, Kuala Lumpur 50603, Malaysia; ar.amani65@gmail.com (A.A.B.); aliyahamizi@um.edu.my (N.A.H.); yasminaw@um.edu.my (Y.A.W.); dr.zaira.chowdhury@um.edu.my (Z.Z.C.)
2 Department of Chemistry, Mutah University, P.O. BOX 7, Mutah, Karak 61710, Jordan; s_alshahateet@mutah.edu.jo
3 Department of Fundamental and Applied Sciences, Universiti Teknologi PETRONAS, Bandar Seri Iskandar 32610, Perak, Malaysia; noorasmawati_mzabidi@utp.edu.my
4 Department of Chemistry and Bioengineering, Vilnius Gediminas Technical University, 10223 Vilnius, Lithuania; m.moosavi1987@gmail.com
5 Department of Industrial and Manufacturing Engineering, University of Wisconsin Milwaukee, Milwaukee, WI 53211, USA; kordija2@uwm.edu
* Correspondence: omid.akbarzadeh63@gmail.com (O.A.); drsureshnano@gmail.com (S.S.)

**Abstract:** The effect of reaction temperature, syngas space velocity, and catalyst stability on Fischer-Tropsch reaction was investigated using a fixed-bed microreactor. Cobalt and Manganese bimetallic catalysts on carbon nanotubes (CNT) support (Co-Mn/CNT) were synthesized via the strong electrostatic adsorption (SEA) method. For testing the performance of the catalyst, Co-Mn/CNT catalysts with four different manganese percentages (0, 5, 10, 15, and 20%) were synthesized. Synthesized catalysts were then analyzed by TEM, FESEM, atomic absorption spectrometry (AAS), and zeta potential sizer. In this study, the temperature was varied from 200 to 280 °C and syngas space velocity was varied from 0.5 to 4.5 L/g.h. Results showed an increasing reaction temperature from 200 °C to 280 °C with reaction pressure of 20 atm, the Space velocity of 2.5 L/h.g and $H_2$/CO ratio of 2, lead to the rise of CO % conversion from 59.5% to 88.2% and an increase for $C_{5+}$ selectivity from 83.2% to 85.8%. When compared to the other catalyst formulation, the catalyst sample with 95% cobalt and 5% manganese on CNT support (95Co5Mn/CNT) performed more stable for 48 h on stream.

**Keywords:** carbon nanotubes; thermal treatment; cobalt; Fischer-Tropsch; catalyst; acid treatment

## 1. Introduction

Fischer-Tropsch Synthesis (FTS) utilizes syngas ($H_2$ + CO) to generate hydrocarbons which have a significant role among eco-friendly fuels and renewable energies. Due to abundant natural gas and coal resources, gas to the liquid process is appealing as a source of feed instead of declining crude oil reservoirs. Fuels produced with FTS are eco-friendly and have very low levels of greenhouse gases. Cobalt catalyst is a popular catalyst choice for FTS [1,2]. It is of most economic interest to have liquid hydrocarbons with long-chain carbon atoms, referred to as $C_{5+}$. We employed the same combined acid and heat pre-treatments of CNT as Tavasoli et al. [3], but the strong electrostatic adsorption (SEA) technique was used for the preparation of the Co/CNT catalyst, with the pH of the precursor solution being regulated throughout the metal deposition. Schwarz proposed that the electrostatic interactions between a metallic ion and a charged support may be used to control the metallic ion's adsorption over surfaces with two oxide fractions [4,5]. The

concept of this methodology has been efficiently employed to generate highly distributed bimetallic catalysts, with a variety of oxide and carbon substrates [6–8]. Depending upon the pH solution is acidic or basic, the hydroxyl (–OH) groups on the surface of an oxide become protonated or deprotonated naturally. These charged hydroxyl groups can then absorb metal complex ions in an oppositely charged solution. The density of charged hydroxyl groups on the oxide surface is determined by the pH at which the surface is neutrally charged, which is called the Point of Zero Charge (PZC). The hydroxyl groups deprotonate above the PZC, making the surface negatively charged and allowing cationic complexes to be adsorbed onto the surface by electrostatic adsorption technique [8]. Previous research on CNT-supported cobalt catalysts is used as an impregnation approach without pH control during catalyst synthesis [9,10]. The deposition of cobalt solution because of pre-treated CNT support has been carried out in this study using the SEA principle at a specific pH. During the synthesis process, the pH has been monitored for cobalt solution. On the characteristics and performance of Co/CNTs catalysts, the impacts of combined acid and heat pre-treatments of CNT support are discussed. The activity and stability of the Co/CNT catalyst in FTS are increased by combining an acid with the thermal pre-treatment of CNT at 900 °C.

Consequently, optimizing the distribution of the reaction product is important and this can be accomplished by varying any of the reaction factors, like temperature, pressure, $H_2/CO$ ratio, reactor type, and catalyst, etc. [11–16]. Increasing FT operating pressure for cobalt-based catalysts was reported to have a negligible impact on enhancing the reaction rate and $C_{5+}$ selectivity [17–19]. As a part of ongoing research, the effect of reaction pressure on Co/CNT catalyst performance with different supporting materials has been studied. The selectivity of shorter molecular hydrocarbons ($C_1$-$C_2$) has been revealed to be substantially enhanced by increasing the reaction temperature and the $H_2/CO$ ratio. However, the selection of long molecular hydrocarbons ($C_{5+}$) is substantially enhanced by reducing the pressure of the reaction [20–23]. Some researchers recorded the influence of operating conditions on the product selectivity for cobalt-based catalysts [20,24,25] and revealed that the olefin selectivity of the hydrocarbon product range reduced with rising pressure, which has reported in prior studies [19,21,24]. This study is a continuation of previous research that has been examined and published [26–30]. The current investigation aims to prepare cobalt manganese bimetallic catalysts on CNT substrate employing SEA technique, also to study effects of temperature, syngas space velocity, and catalyst stability through Fischer-Tropsch synthesis by performing Co-Mn/CNT bimetallic catalyst.

## 2. Process Result Dissection

The effects of temperature, catalyst stability, and syngas space velocity on the catalytic efficiency of monometallic and bimetallic Co-Mn were analyzed. The findings of the reaction were contrasted by product selection in terms of carbon monoxide conversion and hydrocarbon. In the reaction study part, all the reactions were performed two times and the standard deviation value was calculated to be ±1 percent for all reactions. Carbon mass balance was calculated from the moles of carbon entering the reactor relative to the moles of carbonaceous products formed. The advantage and novelty of the current studies were performed by the SEA method for synthesizing Co-Mn catalysts on CNT support for Fischer-Tropsch (FT) reaction, which has not been reported previously [26–29]. A high percentage of CO conversion and $C_{5+}$ selectivity was obtained in the present investigation.

### 2.1. Influence of Reaction Temperature on Catalyst Efficiency

Table 1 revealed that the Fischer-Tropsch synthesis rates, as well as the CO conversion, are under the strong influence of reaction temperature. The results illustrate that rising Fischer-Tropsch reaction temperature from 200 to 280 °C boosts % CO conversion from 59.5 to 88.2%. Increasing FTS temperature rises the motions of hydrogen on the catalyst surface and results in greater CO conversion [31]. Simultaneously, the WGS reaction rate rises from 0.55 to 0.80. The rate of WGS reaction or $CO_2$ formation can be increased and related to the

rise in water semi-pressure, owing to the rise in Fischer-Tropsch synthesis reaction rate [32]. A comparison of hydrocarbon product selectivity for 95Co5Mn/CNT catalysts at 220 °C and 280 °C shows a significant change with decreases in molecular weight hydrocarbons for greater reaction temperature [33]. The results indicate that methane selectivity using 95Co5Mn/CNT catalysts at 200 and 280 °C is 10.8%, and 15.2%, respectively. Further, the selectivity of $C_{5+}$ hydrocarbons for 95Co5Mn/CNT catalysts increases from 83.2% (200 °C) to 85.8% (240 °C). The olefin to paraffin ratio reduced from 0.54 to 0.15. This finding presented that greater reaction temperature enhances the carbon monoxide conversion, but for $C_{5+}$ selectivity, increasing reaction temperature leads to a hydrocarbon chain moves towards a shorter chain [32,34].

**Table 1.** Influence of reaction temperature (°C) on CO conversion%, $C_1$ selectivity%, $C_2$–$C_4$ selectivity%, $C_{5+}$ selectivity%, olefinity and WGS reaction *.

| CO Conversion% | 200 | 220 | 240 | 260 | 280 |
|---|---|---|---|---|---|
| Co/CNT | 48.3 | 54.3 | 58.2 | 59.6 | 59.5 |
| 95Co5Mn/CNT | 59.5 | 78.2 | 86.6 | 87.5 | 88.2 |
| 90Co10Mn/CNT | 57.1 | 73.1 | 79.8 | 80.3 | 81.5 |
| 85Co15Mn/CNT | 55.2 | 67.1 | 73.2 | 74.1 | 74.5 |
| 80Co20Mn/CNT | 50.2 | 61.8 | 66.3 | 67.6 | 67.5 |
| **$C_1$ selectivity%** | | | | | |
| Co/CNT | 15.5 | 16.9 | 16.5 | 18.6 | 19.5 |
| 95Co5Mn/CNT | 10.8 | 11.3 | 11.8 | 13.5 | 15.2 |
| 90Co10Mn/CNT | 12.3 | 12.8 | 13.3 | 14.9 | 16.5 |
| 85Co15Mn/CNT | 13.1 | 13.6 | 14.1 | 15.5 | 17.1 |
| 80Co20Mn/CNT | 14.2 | 14.5 | 15.8 | 16.5 | 18.5 |
| **$C_2$–$C_4$ selectivity%** | | | | | |
| Co/CNT | 12.6 | 13 | 13.4 | 17.3 | 19.6 |
| 95Co5Mn/CNT | 5.5 | 6.1 | 6.7 | 9.6 | 11.3 |
| 90Co10Mn/CNT | 7.7 | 8.2 | 8.7 | 11.8 | 13.7 |
| 85Co15Mn/CNT | 8.5 | 9.8 | 9.4 | 13.6 | 15.6 |
| 80Co20Mn/CNT | 9.6 | 10.6 | 10.5 | 14.2 | 16.5 |
| **$C_{5+}$ selectivity%** | | | | | |
| Co/CNT | 72.1 | 71.1 | 69.1 | 56.7 | 50.6 |
| 95Co5Mn/CNT | 83.2 | 82.4 | 85.8 | 68.2 | 61.7 |
| 90Co10Mn/CNT | 82.5 | 81.6 | 78.7 | 67.3 | 60.6 |
| 85Co15Mn/CNT | 80.5 | 79.5 | 76.5 | 64.4 | 58.4 |
| 80Co20Mn/CNT | 78.5 | 77.5 | 74.5 | 63.5 | 55.2 |
| **Olefinity** | | | | | |
| Co/CNT | 0.92 | 0.71 | 0.63 | 0.72 | 0.84 |
| 95Co5Mn/CNT | 0.54 | 0.19 | 0.15 | 0.22 | 0.38 |
| 90Co10Mn/CNT | 0.63 | 0.34 | 0.27 | 0.37 | 0.55 |
| 85Co15Mn/CNT | 0.78 | 0.42 | 0.34 | 0.5 | 0.65 |
| 80Co20Mn/CNT | 0.87 | 0.57 | 0.45 | 0.58 | 0.75 |
| **WGS selectivity** | | | | | |
| Co/CNT | 0.24 | 0.33 | 0.36 | 0.45 | 0.53 |
| 95Co5Mn/CNT | 0.55 | 0.58 | 0.65 | 0.76 | 0.80 |
| 90Co10Mn/CNT | 0.48 | 0.51 | 0.58 | 0.68 | 0.74 |
| 85Co15Mn/CNT | 0.43 | 0.45 | 0.51 | 0.61 | 0.69 |
| 80Co20Mn/CNT | 0.38 | 0.43 | 0.45 | 0.56 | 0.64 |

* Reaction condition: Pressure: 20 atm, Space velocity: 2.5 L/h.g, Ratio of $H_2$/CO:2.

Table 1 illustrates five reaction temperatures: 200, 220, 240, 260, and 280 °C, which were used at an $H_2$/CO feed proportion of two and a pressure of 20 atm on different Co-Mn/CNT catalyst compositions. As seen in Table 1, increasing the reaction temperature enhanced carbon monoxide conversion. It has been shown that increasing the temperature

promotes CO molecule dissociation on active sites of catalysts, hence, increasing the rate of CO hydrogenation [35]. According to Vannice and coworkers [36], CO molecules become active at higher operating temperatures due to the strong interaction of C and O atoms with active metal surfaces [37]. However, the reaction temperature has been increased to 260 °C, activity dropped again, indicating that the optimum reaction temperature was 240 °C. Diffusion effects have been suggested by many studies for an increase in the activity of FTS catalysts as temperature rises [38]. It has been claimed that when the operating temperature rises, the migration of molecules away from active sites improves, increasing the number of active sites available. These findings were comparable to those of prior studies [37].

The methane is used as a fuel, the production of $C_1$ and $C_2$–$C_4$ should be reduced to a minimum during FTS [39]. As indicated in Table 1, increasing the reaction temperature improved methane selectivity. The increase in methane selectivity with increasing temperature has been attributed to an increase in CO molecule dissociation on the catalyst surface, resulting in more carbons available for hydrogenation by $H_2$ molecules [40].

$C_{5+}$ selectivity is considered a preferred FTS output, and reaction conditions are geared toward increasing $C_{5+}$ products. Table 1 summarizes the effects of temperature on $C_{5+}$ selectivity. With a drop-in temperature, selectivity for $C_{5+}$ increased. During the FTS process, it has been found that increasing the operating temperature reduces chain propagation and improves the chain termination step [41]. The FTS is a surface polymerization reaction, increasing the temperature reduces selectivity for long-chained molecules while improving selectivity for lower hydrocarbons.

As indicated in Table 1, increasing temperature resulted in a decrease in olefinity. In the temperature range of 220 to 250 °C, the Olefin to Paraffin ratio declined faster than in the temperature range of 250 to 280 °C, when it climbed again, which was consistent with thermodynamic assumptions. Olefins are generated first, then propagated to form long-chained hydrocarbons during FTS. As most olefins were hydrogenated, an increase in temperature increased CO hydrogenation and hence decreased olefinity [42]. CO hydrogenation decreased at the temperature range of 250 to 280 °C due to the high chemisorption of CO molecules on the catalyst surface, which reduced the likelihood of hydrogen molecules hydrogenating CO molecules. Hunter and coworkers [42] noticed a similar pattern. On the other hand, some researchers discovered a contrary tendency, claiming that increasing the operating temperature increased olefinity. The rate of WGS increased with increasing temperature for monometallic and bimetallic catalysts (Table 1), and other researchers have found the same pattern [42].

*2.2. Influence of Space Velocity on Catalyst Efficiency*

Effect of mass space velocity (Vm) on the catalytic efficiency of CNT-supported monometallic and bimetallic catalysts of Co and Mn have selected at 240 °C with a total flow rate of reactants varied between 0.5, 1.5, 2.5, 3.5, 4.5 L/g.h with $H_2$/CO feed ratio of 2/1 and reaction pressure of 20 atm. The results in Table 2 show that for Co-Mn/CNT, as space velocity increased, CO conversion decreased. Liu and co-workers [43] reported a similar trend for commercial Co-Mn catalysts where it was observed that conversion of CO was lowered to 30% from 82 % by an increase in space velocity from 0.46 to 1.85 L/g.h. Similar results have been previously reported [44]. Catalyst's weight was kept constant, which shows that space velocity was influenced by the total flow rate of reactants. Consequently, the variations in the selectivity of the product would be because of the residence time.

**Table 2.** Effect of space velocity (L/g.h) on CO conversion%, $CH_4$ selectivity% and $C_{5+}$ selectivity% *.

| CO Conversion% | 0.5 | 1.5 | 2.5 | 3.5 | 4.5 |
|---|---|---|---|---|---|
| Co/CNT | 72.2 | 71.3 | 69.7 | 65.8 | 50.7 |
| 95Co5Mn/CNT | 89.7 | 88.9 | 86.6 | 82.0 | 67.4 |
| 90Co10Mn/CNT | 82.4 | 81.6 | 78.4 | 72.7 | 60.4 |

**Table 2.** *Cont.*

| CO Conversion% | 0.5 | 1.5 | 2.5 | 3.5 | 4.5 |
|---|---|---|---|---|---|
| 85Co15Mn/CNT | 80.9 | 79.7 | 76.6 | 71.6 | 58.6 |
| 80Co20Mn/CNT | 78.3 | 77.5 | 74.8 | 70.3 | 55.8 |
| **$C_1$ Selectivity%** | | | | | |
| Co/CNT | 12.6 | 13.5 | 13.4 | 16.3 | 19.4 |
| 95Co5Mn/CNT | 5.5 | 6.1 | 6.7 | 9.7 | 11.8 |
| 90Co10Mn/CNT | 7.7 | 8.2 | 8.7 | 11.9 | 13.6 |
| 85Co15Mn/CNT | 8.5 | 9.7 | 9.4 | 12.5 | 15.7 |
| 80Co20Mn/CNT | 9.6 | 10.6 | 10.5 | 14.4 | 16.5 |
| **$C_{5+}$ Selectivity%** | | | | | |
| Co/CNT | 72.1 | 71.1 | 69.1 | 63.5 | 50.6 |
| 95Co5Mn/CNT | 88.5 | 87.7 | 85.8 | 72.3 | 61.7 |
| 90Co10Mn/CNT | 82.9 | 81.6 | 78.5 | 70.5 | 60.6 |
| 85Co15Mn/CNT | 80.5 | 79.5 | 76.5 | 68.4 | 58.4 |
| 80Co20Mn/CNT | 78.5 | 77.5 | 74.5 | 66.7 | 55.7 |

\* Reaction condition: Pressure: 20 atm, Temperature: 240 °C, Ratio of $H_2$/CO: 2.

## 3. Catalyst Stability and Used Catalyst TEM

Figure 1 shows carbon monoxide conversion with time on stream (TOS) for as-received Co/CNT, 95Co5Mn/CNT, 90Co10Mn/CNT, 85Co15Mn/CNT, and 80Co20Mn/CNT catalysts samples. Catalysts showed different stability patterns within a period of 48 h.

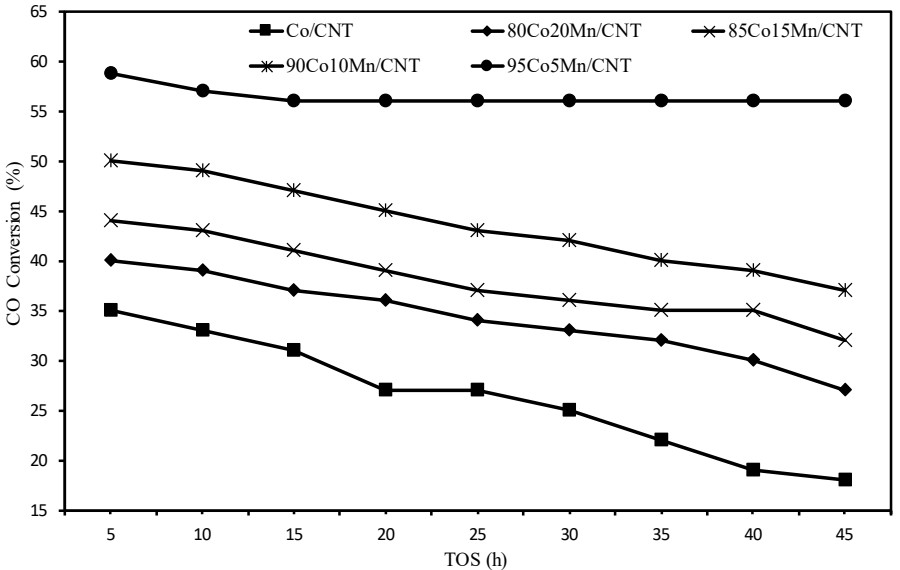

**Figure 1.** Time on stream (TOS) efficiency for catalysts deposited on CNT pre-treated at 240 °C, 20 atm, and $H_2$/CO = 2 reaction conditions.

Figure 1 shows the stability of 95Co5Mn/CNT catalyst in contrast to other formulation catalyst samples. For Co/CNT catalyst sample, CO conversion dropped drastically from 35 to 18% during 48 h. For 95Co15Mn/CNT catalyst results show a slow deactivation from 58.7% of carbon monoxide conversion to 56.9% within 48 h. The stability of catalysts may be related to Mn%, functional groups, structure, defects, and morphology of CNT substrate [33]. For catalyst prepared on 95Co5Mn/CNT pre-treated at 900 °C, CO conversion and $C_{5+}$ selectivity were determined as 58.7%, and 59.1%, respectively. The superior efficiency of 95Co5Mn/CNT compared to other catalyst samples attributed to the higher dispersion and reducibility of cobalt-oxide nanoparticles were confined inside the CNT channels [45]. Figure 2 depicts the TEM images of the catalysts at temperatures of (a)

600 and (b) 900 °C. The particle size was found to be raised from 4.2 to 20.5 nm at 600 °C whereas 7.2 to 14.1 nm at 900 °C to indicate the treated catalyst samples [46].

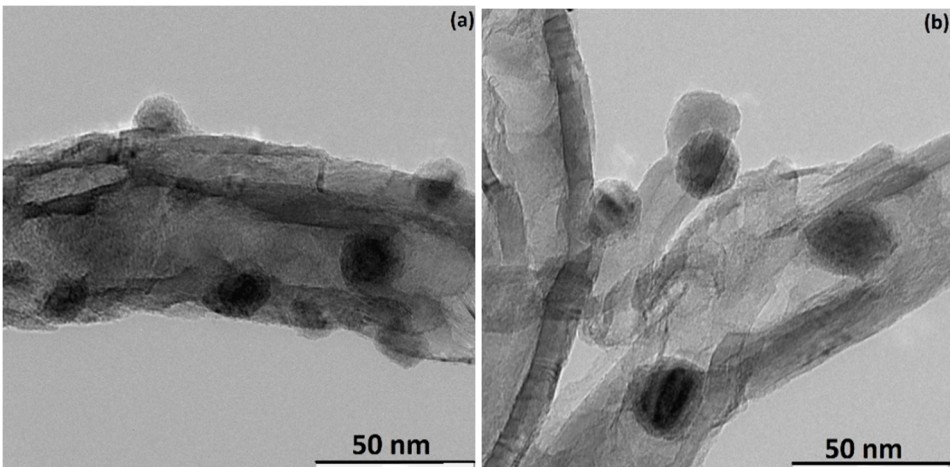

**Figure 2.** TEM image of the catalyst after FTS with thermal treatment (**a**) 600 °C (**b**) 900 °C.

Catalyst deactivation reveals that sintering was extremely high during FTS. The results of the TEM revealed that the active sites on the outside surface of the CNT sinter at a faster rate than the active sites inside the CNT channels. As previously stated, the majority of the cobalt active sites are enclosed within CNT.

The confinement of the reaction liquid inside the pores can improve their interaction, allowing more cobalt active sites to be exposed, thus, encouraging the growth of longer hydrocarbon chains [47]. The findings are consistent with those of other researchers [48], who found that the inside surface of CNT has an electron deficiency, which can enhance CO separation and lead to the synthesis of longer hydrocarbon chains [49]. The catalyst heated at 900 °C resulted in high nanoparticles in the channels with the decreased deactivation rate [50], according to our TEM results (Table 3), increasing the ratio of active sites enclosed inside CNT channels to active sites outside of CNT channels is thought to be a major component in improving $C_{5+}$ selectivity and lowering $CH_4$ rates [51]. The difference in electron dispersion between the internal and external surfaces of the CNT, as well as the cobalt particle confinement phenomena [48]. Due to the electron deficit on the inner surface of the CNT, there is a strong interaction between cobalt oxides and the support. Since the lower sintering potential when compared to the catalyst active sites on the external surface of the CNT channel.

**Table 3.** Textural properties of Co/CNT catalysts at various wt% loading.

| Samples | BET Surface Area (m$^2$/g) | Total Pore Volume (m$^3$/g) |
|---|---|---|
| Pristine CNT | 138.2 | 1.58 |
| CNT.A | 223.2 | 0.88 |
| CNT.A.T | 266.4 | 0.54 |
| Co/CNTs.A | 198.5 | 0.55 |
| 95Co5Mn/CNT.A.T | 217.5 | 0.36 |
| 90Co10Mn/CNT.A.T | 220.8 | 0.48 |
| 85Co15Mn/CNT.A.T | 223.4 | 0.55 |
| 80Co20Mn/CNT.A.T | 225.3 | 0.58 |

## 4. Experimental

### 4.1. Functionalization of CNT Substrate

Functionalization and activation by introducing functional groups to the CNT support using nitric acid are essential before metal loading [49]. The functionalization course

was aimed at improving the interaction among catalyst active sites and the CNT substrate surface. Pre-treatment with acid purifies synthesized CNT, adds oxygen-containing functional groups (–OH) on the catalyst support surface, and removes the fullerene cap from carbon nanotubes to have open CNT channels [50]. A wet chemical oxidation method is the commonly accepted process for activating and functionalizing carbon nanotubes. Around 2 g of purchased CNT (purity > 95%, CVD, length: 10−20 μm, diameter: 30−50 nm, Nanostructured and Amorphous Materials Inc.) were added to a single necked round bottom flask and add 35 vol% nitric acid (Merck) at 110 °C for 10 h [51]. After reflux, the blend was cooled down to ambient temperature, diluted with deionized water, filtered using a filter membrane of 0.2 μm pore size, and washed many times till the residue filtrate pH reached about 7 [52]. Neutralized CNT was dried overnight in an oven at 120 °C and acid-treated CNT continued with thermal treatment for 3 h at 900 °C under flowing argon gas at 20 mL min$^{-1}$ [53].

### 4.2. Point of Zero Charges (PZC), Co Adsorption on CNT, and Catalyst Preparation

The common technique of impregnation was used to synthesize cobalt catalysts which produced a heterogeneous distribution of cobalt catalyst active sites on the substrate, but the Strong Electrostatic Adsorption (SEA) technique lead to greater catalyst particle dispersion and narrower distribution of catalyst size [54–57]. CNT, silica, alumina, and other metal oxides supports have hydroxyl groups on the surface. Based on the SEA technique, the point of zero charges is the pH value of the medium that the hydroxyl group on the surface remains neutral. A range of tests was carried out to find the optimum of catalyst metal active sites on CNT substrate by utilizing cationic hexamine of complexes of catalyst metal. Graph pattern shows metal adsorption increases meaningfully at pH > PZC [54,56,57]. Catalyst samples made via the SEA process [58–62] at optimal pH were found with lower particle size and higher dispersion in contrast to catalyst samples synthesized by the common impregnation technique.

Equilibrium pH at high oxide loading (EpHL) technique [58] was conducted to find the PZC of CNT substrate. The pH value was adjusted range of 2–14 by the addition of nitric acid or ammonium hydroxide to distilled water. By pouring into a conical flask, 0.5 g weighted CNT was added up with the addition of 50 mL of each solution. A rotary shaker was used to shake the mixture for 1 h before measuring the final pH value. Figure 3a performed PZC of CNT support at pH 9.5. The pH of the cobalt nitrate precursor solution was set to a range of 2–14 to study the cobalt adsorption against pH. Weighed CNT was combined into solutions and shaken for 1 h, and the final pH was then measured. The volume of 5ml of filtered cobalt solution of every sample was analyzed for the percentage of cobalt via atomic absorption spectrophotometer (AAS). Figure 3b indicates the plot of Co adsorption versus pH and showed optimal pH for Co adsorption is 14. At chosen pH = 14, the cobalt precursor was uptake by 10 wt% at Co-Mn metal loads from an excess solution on CNT substrate to avoid pH change. The sample was filtered and dried for 24 h under airflow. The dried sample was calcinated in a tubular furnace at 400 °C for 4 h under airflow to eliminate residual reactants.

Based on the SEA preparation method, the surface of functionalized CNT changed to negatively charged, the pH solution was greater than the PZC of the CNT. The PZC of the CNT support was found to be 9.5. The highest cobalt adsorbed on the CNT happened when the Co precursor solution remained at a pH of 14. Accordingly, the uptake of Co ions on the pre-treated CNT was occurred at pH 14 using a solution of $Co(NO_3)_2$. Dried catalyst samples were calcined in a tubular furnace at 400 °C for 4 h under Ar gas flow. The metal loading on CNT was performed at 10 wt% during the catalyst synthesis period.

### 4.3. Catalyst Characterization

Fischer-Tropsch catalyst performance is significantly affected by catalyst physicochemical properties. Therefore, it is important to characterize the catalyst's physicochemical characteristics. FTS catalyst surface physical and chemical properties, such as catalytic ac-

tivity and selectivity were characterized. Figure 4 shows Transmission Electron Microscopy (TEM) of different catalyst samples conducted by a Zeiss LIBRA 200 FE TEM at 200 kV accelerating voltage. TEM results presenting catalyst samples with 5% Mn have the highest dispersion and narrow size metal particle size distribution. The rise in the Mn metal % from 5 to 20%, lead to enhance the catalyst active sites adsorption on the CNT substrate and particularly increasing from 15 and 20%, agglomeration phenomena of catalyst particles occurred and catalyst active sites agglomerate on CNT support and lead to a decline of catalyst CO conversion up to 25% and $C_{5+}$ selectivity up to 10%.

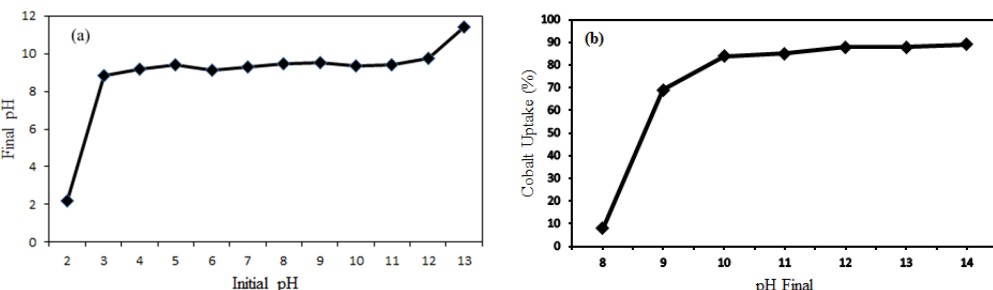

**Figure 3.** The finding of (**a**) PZC of CNT support, (**b**) Co-Mn adsorption versus pH survey by AAS.

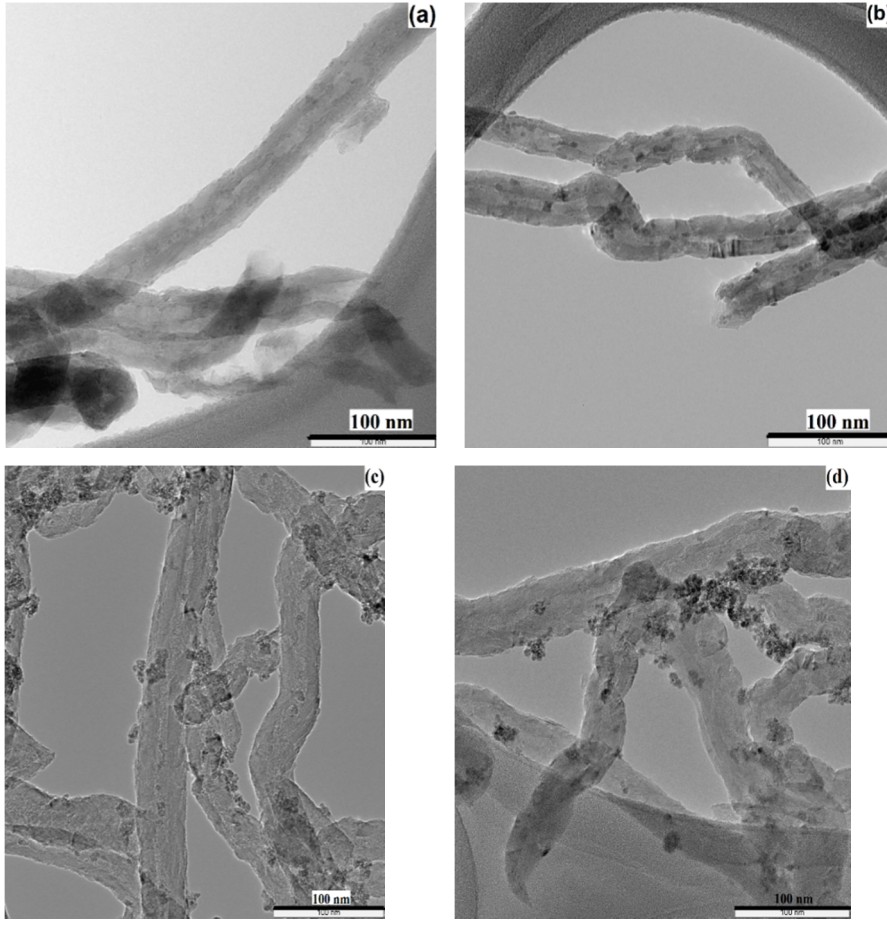

**Figure 4.** TEM images of (**a**) 95Co5Mn/CNT, (**b**) 90Co10Mn/CNT, (**c**) 85Co15Mn/CNT, (**d**) 80Co20Mn/CNT catalysts.

Field-Emission Scanning electron microscopy (FESEM) was used to evaluate sample morphology and elemental surface structure using a Zeiss Supra 55 VP with voltage acceleration: 5 KV, magnification: 100.00 KX, and operating distance: 4 mm. (Figure 5 FESEM images confirm and support the TEM results, demonstrating that increasing the

Mn percent from 5 to 20%, catalyst active sites agglomerate on CNT support, and lead to a decline of catalyst CO conversion and $C_{5+}$ selectivity up to 25%, and 10%, respectively. Atomic Absorption Spectrometer (AAS) was used by Agilent Technologies GTA 120 to evaluate cobalt and manganese adsorption on the CNT substrate.

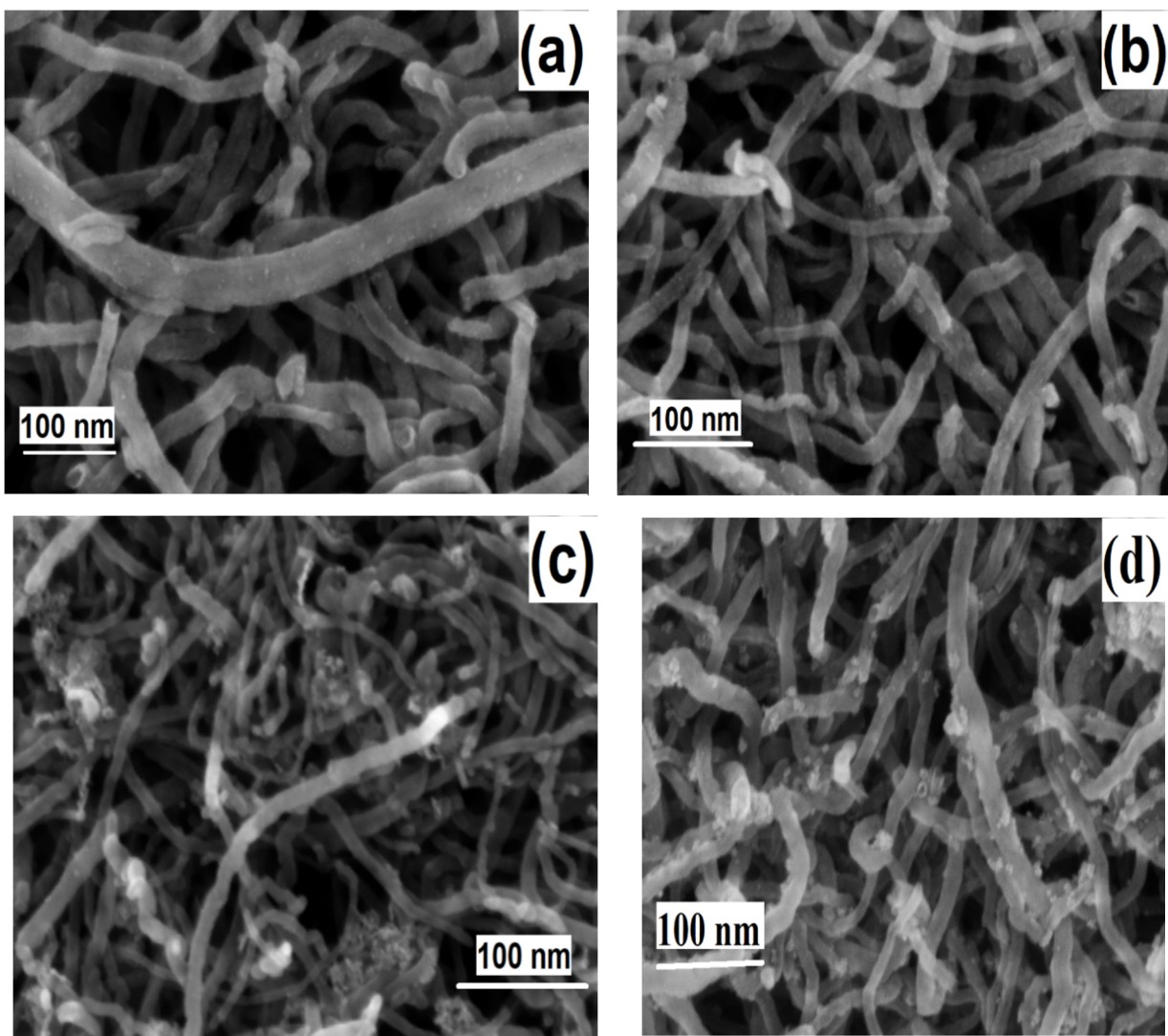

**Figure 5.** FESEM images of (**a**) 95Co5Mn/CNT, (**b**) 90Co10Mn/CNT, (**c**) 85Co15Mn/CNT, (**d**) 80Co20Mn/CNT catalysts.

### 4.4. Reactor Setup, Product Sampling, and Analysis

FTS performed in a continuous flow fixed-bed with the Micro-activity-reference reactor (Micromeritics, Norcross, GA, USA) were attached with mass flow controllers (Hi-Tec Bronkhorst, Ruurlo, The Netherlands). Carbon monoxide and $H_2$ were applied as reactant gases. The amount of 0.02 g catalyst was located in a stainless-steel reactor chamber (9 mm i.d. $\times$ 200 mm length) and placed in quartz tools without any dilution. Prior to the reaction, the catalysts were lowered in-situ beneath $H_2$ flow at 0.1 MPa and 420 °C for 10 h. The process was performed in different reaction parameters for 48 h time-on-stream (TOS). The reactor was attached to the gas chromatograph (Agilent Hewlett-Packard Series 6890, Santa Clara, CA, USA) attached with two TCD and one FID detector. Products were analyzed every 30 minutes using DB-5 column. Hydrocarbon selectivity (FID1: Methane, Ethane, Propane, Ethylene, Iso-butane, n-butane, n-pentane, n-hexane, n-heptane, TCD2: $CO_2$, CO, $N_2$, $O_2$, and TCD3: $H_2$) were calculated after reaction completion (10 h). The results were collected at a steady-state setup using a carbon balance of 99–102%. The reproducibility

was checked by doing all reactions two times under the same reaction and catalyst terms. STD of experimental results were ±5.0%. The CO, methane ($CH_4$), and $C_{5+}$ selectivity conversion percentages were analyzed using Equations (1)–(3) respectively [63]:

$$\text{CO conversion } (\%) = \frac{CO_{in} - CO_{out}}{CO_{in}} \times 100 \tag{1}$$

$$CH_4 \text{selectivity}(\%) = \frac{\text{Mole of } CH_4}{\text{Total moles of hydrocarbons}} \times 100 \tag{2}$$

$$C_{5+} \text{selectivity}(\%) = \frac{\text{Moles of } C_{5+}}{\text{Total moles of hydrocarbons}} \times 100 \tag{3}$$

The FTS level shown in Equation (1) and the reaction rate of the water gas change (Equation (5)) is equal to the carbon dioxide formation rate ($RFCO_2$) and can be described by [60,64,65]:

$$RFTS(g \ HC/gcat/h) = g \ \text{hydrocarbons produced}/gcat * h^{-1} \tag{4}$$

$$RWGS(gCO_2/gcat/h) = RFCO_2 = gCO_2 \text{produced}/gcat * h^{-1} \tag{5}$$

It is a significant step to ease and initiate calcined catalysts before reaction. Catalysts were reduced to 12.5 h at 420 °C under 1.8 L/g.h flow of $H_2$. After catalyst in-situ activation, the temperature was reduced to the required temperature of the Fischer-Tropsch reaction, and the reactor tube flushed for 10 min with helium gas. Fischer-Tropsch reaction was carried out at 2/1 $H_2$/CO (*v/v*) ratio and 20 atm pressure. Additional experiments were performed to explore the impacts of space velocity (0.5, 1.5, 2.5, 3.5, and 4.5 L/g.h), temperature (200, 220, 240, 260, 280 °C), and catalyst stability by conducting different catalysts. Figure 6 shows the schematic diagram of the micro activity-reference reactor (Micromeritics).

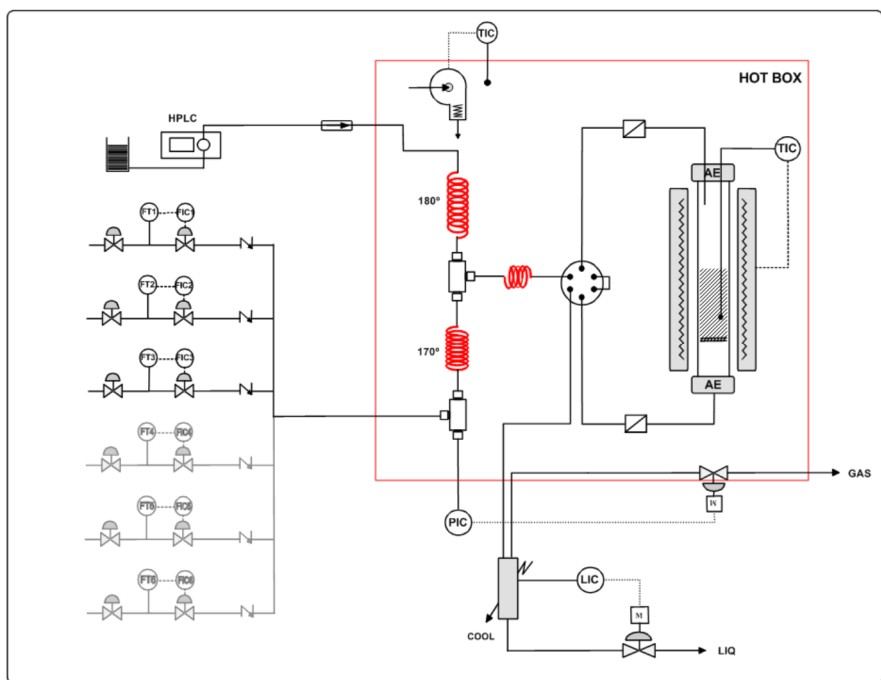

**Figure 6.** Schematic diagram of the micro activity-reference reactor (Micromeritics).

The textural properties of the BET surface area and total pore volume are shown in Table 3. According to the results, the total area (BET) increased from 217.5 to 225.3 m$^2$/g with a 5 to 20% increase in Mn load. Higher nanoparticle dispersion may be causing an increase in surface area. From the findings, the overall pore volume increased from 0.36 to

0.58 (m$^3$/g) as the Mn percent of catalysts raised from 5% to 20%. The addition of cobalt and manganese to CNT support increased overall pore volumes in both BET surface areas.

The XRD patterns of CNT support and catalyst samples are shown in Figure 7. The peaks at 26° and 44° correspond to carbon nanotubes [66]. Diffraction peaks of $Co_3O_4$ spinel appear in the monometallic Co/CNT sample in the ranges of 32°and 37.1° [66]. At two values of 32.5° and 44°, the A.T sample reveals a hematite pattern ($Mn_2O_3$) [67]. $Co_3O_4$ spinel diffraction peaks were observed at 32.5 and 37.1° in bimetallic 95Co5Mn/CNT catalyst XRD patterns. Due to the low manganese content in the catalyst, $Mn_2O_3$ was only linked with a weak peak at 44°.

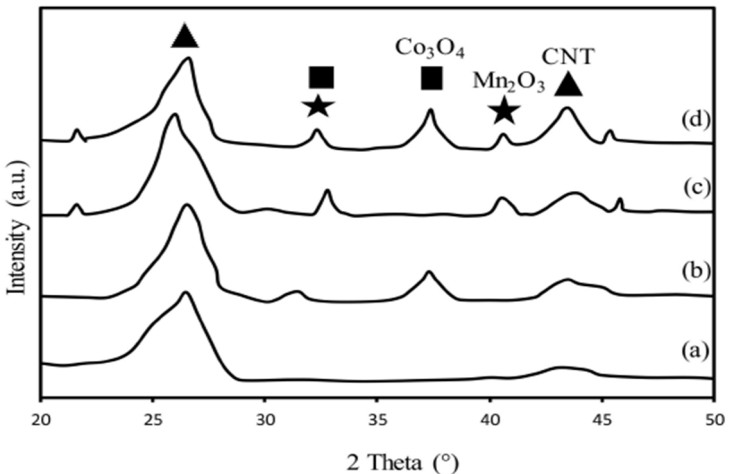

**Figure 7.** XRD patterns of (**a**) CNT (**b**) Co/CNT. A.T (**c**) Mn/CNT.A.T (**d**) 95Co5Mn/CNT.A.T catalysts.

Figure 8 shows XRD patterns of calcined catalysts with various Mn metal loading. The peak at 25° and 43° shows unique $Co_3O_4$ crystal planes [68]. For $Co_3O_4$, the most significant peak was seen at 36.8°. Lower intensity peaks were detected at 32.5° and 44°, showing Mn oxide diffraction peaks, due to the limited number of Mn promoters in the catalyst XRD pattern. The average particle size of the catalysts was estimated as 6-8 nm using XRD and TEM images [69–75]. Table 3 shows that as manganese load increases from 5% to 20%, the average particle size of $Co_3O_4$ drops from 7.5 to 6.5 nm, which is similar to the results of the TEM study (Figure 4). The agglomeration of cobalt particles raises the average particle size. The average particle size drops somewhat when Mn is added to the Co catalyst, as seen in Table 3.

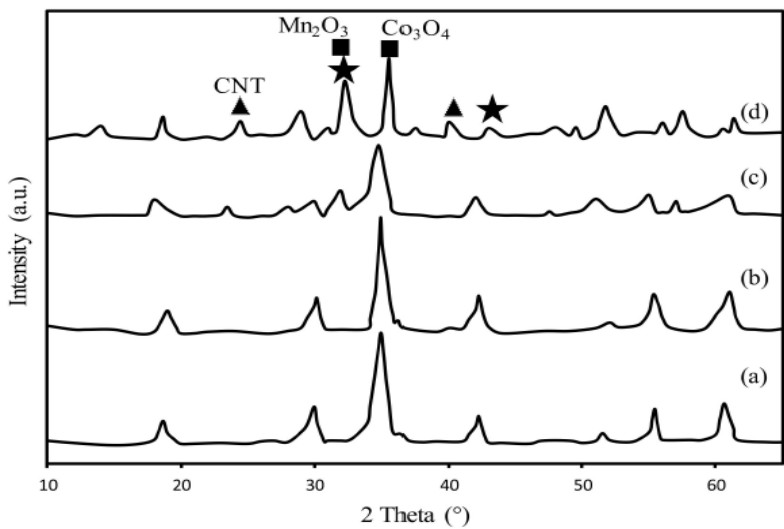

**Figure 8.** XRD profile of Co–Mn/CNT with Mn content (**a**) 5, (**b**) 10, (**c**) 15 and (**d**) 20%.

## 5. Conclusions

The Cobalt-Manganese bimetallic catalyst was synthesized by acid and thermal-treated CNT substrate using the SEA process. The efficiency of various percentage formulations of the Co-Mn catalyst supported on CNT was verified by the FTS reaction. High stability was proven by 95Co5Mn/CNT catalyst for more than 45 h. It was concluded that reaction variables created a high impact on catalytic activities and product selectivities during the FTS process. An increase in reaction temperature up to 280 °C enhanced carbon monoxide percent conversion up to 88.2% and reduced $C_{5+}$ selectivity up to 55.2%, while increased WGS rate up to 0.8. An increase in space velocity up to 4.5 (L/g.h) decreases CO percent conversion to 55.8% and decreases $C_{5+}$ selectivity to 55.7%. However, after optimization analysis, 95Co5Mn/CNT catalyst formulation showed a high efficiency at 240 °C with a space velocity of 2.5(L/g.h). In the mentioned condition, carbon monoxide conversion and $C_{5+}$ selectivity were 86.6% and 85.8% respectively.

**Author Contributions:** Data curation, N.A.M.Z., S.M., A.A.B., N.A.H. and Z.Z.C.; Formal analysis, N.A.M.Z., S.M., A.K., Z.Z.C. and S.S.; Funding acquisition, S.F.A.; Investigation, A.K., Y.A.W. and S.S.; Validation, N.A.H. and Y.A.W.; Visualization, A.K., A.A.B., N.A.H. and Y.A.W.; Writing—original draft, O.A.; Writing—review & editing, S.S. All authors have read and agreed to the published version of the manuscript.

**Funding:** This research was funded by the Ministry of Education, Malaysia under the Fundamental Research Grant Scheme (FRGS/1/2012/SG01/UTP/02/01).

**Data Availability Statement:** All data generated or analysed during this study are included in this published article.

**Acknowledgments:** The authors acknowledge the Universiti Teknologi PETRONAS and University of Malaya.

**Conflicts of Interest:** The authors declare no conflict of interest with this work.

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
