# Peer review of "Effect of Temperature, Syngas Space Velocity and Catalyst Stability of Co-Mn/CNT Bimetallic Catalyst on Fischer Tropsch Synthesis Performance"

_catalysts, doi:10.3390/catal11070846_

Round 1

Reviewer 1 Report

The manuscript presents the studies of Co-Mn/CNT bimetallic catalysts for Fischer-Tropsch synthesis. The effects of temperature, syngas space velocity and catalyst’s stability were analyzed and discussed. In general, the presented studies are interesting and well done.  The activation effect of manganese addition was clearly shown. However, I would like to suggest some corrections of the manuscript before its acceptance:

  • The physio-chemical characteristic of the studied catalysts was presented in previous paper. In my opinion summary characteristic of these studied catalysts should be also presented in this manuscript, e.g., in the form of table presenting at least real content of Co and Mn in the samples, their specific surface area and pore volume, Co- and Mn-containing phases identified by XRD.
  • The catalytic performance of the studied samples should be discussed with relation to the data listed in table from point 1.
  • The results of the stability tests are very interesting (Fig. 5). Have authors verified reproducibility of such test for the most promising (stable) catalyst (95Co5Mn/CNT)?
  • The results presented in the manuscript should be compared and discussed with other catalytically active systems for this reaction, reported in scientific literature.

Author Response

Thanks for the comments given by the referees. These comments helped us to improve the quality of our paper.

Furnished below is the pointwise reply for the comments of the Referees and the actions incorporated in the manuscript.

Reviewer 2 Report

The article entitled "Effect of Temperature, Syngas Space Velocity and Catalyst Stability on Co-Mn/CNT Bimetallic Catalyst Performance" has been reviewed and in my opinion, is not of sufficient quality to be accepted for publication in Catalysts.

After reading the manuscript I noticed that the authors were not very clear in their exposition, namely in the introduction that is very weak regarding the importance and novelty of this work concerning the previous research. The state of the art is also scarce.

The experimental details regarding the preparation of the catalysts are insufficient and regarding the characterization of the catalysts, the authors refer to the techniques used without discussing them and relating them to the catalytic activity. The importance and effect on the catalytic performance of this particular support are also not mentioned. Figure 3 needs to be resized.

There are many sentences that seem to lack continuation. For example in line 234 the authors say "C5+ products are desired FTS products" but what are the products they obtained? And selectivity? 

The authors present in section 4 the "Catalyst stability" but only present the results of the reuse. What about the characterization of the catalysts after catalysis? 

In general, the manuscript has to be much improved to be accepted for publication.

Author Response

(The authors gave the same response as above.)

Reviewer 3 Report

Cobalt and manganese bimetallic catalysts on CNT support have been obtained by impregnation method and studied in Fischer-Tropsch synthesis. The work may be of interest for the readers. In general, the manuscript is written in a clear and easy-to-follow style. The experiments are appropriate for catalyst characterization, however, the description and interpretation of the results requires a throughout revision before acceptance for publication.

Specifuc comments:

  1. Title needs revision to mention Fischer-Tropsch synthesis
  2. The abstract needs to be made more informative. Abbreviations should not be used. For example, instead of sample code “5Co5Mn/CNT” it should be 5 wt% Co 5wt.%Mn/CNT in the abstract. Also experimental conditions such as reactor type, gas composition, pressure and space velocity range/residence time should be mentioned.
  3. The details of GC analysis should be provided. Which column was used, how many products were analyzed. Could you mention these products explicitly?
  4. Could you present separately the data for saturated and unsaturated hydrocarbons. This is important as you use terminology “olefinity” but there are no supporting data.
  5. The absence of thermal gradients in the reactor and in the catalyst pellet should be supported by calculations
  6. For discussion on data from Table 1&2. Could you calculate the reaction rate (in mol/g-s) and compare it with literature data over similar catalysts?
  7. Like 243. During FTS, olefins are formed first and then propagated to form long-chained hydrocarbons. Add a reference.
  8. Line 271. Similar trends have been reported earlier [55]. Could you make a more quantitative comparison. Mention experimental conditions in both studies for proper comparison. What was the same and what was different in these studies?
  9. Could you comment on the accuracy of your data. Did you repeat each measurement? You provided results with three significant digits, for example line 296, a selectivity of 58.7% is reported. However the carbon balance was reported to be (line 158) within 7% (95–102%).

Author Response

(The authors gave the same response as above.)

Round 2

Reviewer 2 Report

The authors have improved the introduction, however, I still feel that the relation between catalytic performance and catalyst morphology is not sufficiently explored.

The authors claim to have identified several products namely Methane, Ethane, Propane, Ethylene, Acetylene, Iso-butane, n-butane, n-pentane, n-hexane, n-heptane, CO2, CO, N2, and O2 but only refer to gas chromatography technique. Some of the products seem to be difficult to identify by this technique. I challenge the authors to attach a chromatogram in the supplementary material with the identification of all products.

Figure numbering is wrong.

I can't understand why the authors marked in blue a large part of the text of the original version, looking like there was a big change in the manuscript when in fact this doesn't happen.

I still feel that the manuscript is of relatively low quality and precision, and so I leave the decision to the editor. 

Author Response

(The authors gave the same response as above.)

Reviewer 3 Report

Most of comments have been answered, thanks. However there are a few more related to the new text added to the revised version.

1) Line 336 “The amount of 0.02 g catalyst was located in a stainless-steel reactor chamber (9 mm i.d. x 200 mm length)”. The catalyst particle size is not provided as well as the question about the absence of flow limitations needs to be addressed. In other words, could the authors demonstrate that the reaction rate does not depend on the flow rate? Just present two or three experiments with different flow rates and calculate the reaction rate. 

2) Line 343. Analysis section has been added but remains unlear. It is stated that all gases (Methane, Ethane, Propane, Ethylene, Acetylene, Iso-butane, n-butane, n-pentane, n-hexane, n-heptane, CO2, CO, N2 and O2) were separated using DB-5 column. This does not look realistic. Could you show example of separation of N2 and O2 with this column?  Also could the authors specify how CO2 was analyzed? This is not possible with FID. Please provide parameters of the column (length, film thickness). This information is very important to conclude on the accuracy of the data provided.

3) Line 345. Now a carbon balance is reported to be of 99–102%. In the original version it was reported to be 95–102%. What has been done to improve it by 4%. Could you add a data file in supplementary information to demonstrate the concentrations of all products?

4) Line 346. “The reproducibility was checked by doing all reactions two times under the same reaction and catalyst terms”

However line 84 reports “or the reaction study part, all the reactions were performed 84 three times and the standard deviation value was ±1 percent for all reactions” This is confusing and needs to be corrected.

5) English needs to be checked in many places.

Author Response

(The authors gave the same response as above.)

Round 3

Reviewer 2 Report

The authors have improved the manuscript and therefore I suggest that it be accepted for publication.